# Systematic Review of English/Arabic Machine Translation Postediting: Implications for AI Application in Translation Research and Pedagogy

**Lamis Ismail Omar** * 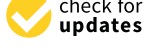 **and Abdelrahman Abdalla Salih**

Department of English Language and Literature, College of Arts and Applied Sciences, Dhofar University, Salalah 211, Dhofar, Oman; asalihahmed@du.edu.om
* Correspondence: lomar@du.edu.om

**Abstract:** The twenty-first century has witnessed an extensive evolution in translation practice thanks to the accelerated progress in machine translation tools and software. With the increased scalability and availability of machine translation software empowered by artificial intelligence, translation students and practitioners have continued to show an unwavering reliance on automatic translation systems. Academically, there is little recognition of the need to develop machine translation skillsets amongst translation learners in English/Arabic translation programs. This study provides a systematic review of machine translation postediting with reference to English/Arabic machine translation. Using the Preferred Reporting Items for Systematic Review and Meta-Analysis, the paper reviewed 60 studies conducted since the beginning of the twenty-first century and classified them by different metrics to identify relevant trends and research gaps. The results showed that research on the topic has been primarily prescriptive, concentrating on evaluating and developing machine translation software while neglecting aspects related to translators' skillsets and competencies. The paper highlights the significance of postediting as an important digital literacy to be developed among Arabic translation students and the need to bridge the existing research and pedagogic gap in MT education.

**Keywords:** machine translation; artificial intelligence; postediting; English/Arabic translation; COVID-19; translation pedagogy

## 1. Background

### 1.1. MT Evolution

Ever since Machine Translation (MT) emerged in the mid-1950s [1], diverse aspects of automatic translation systems have witnessed groundbreaking evolution, leaving a durable impact on educational and professional fields worldwide, including on foreign language education and translation education [2–5]. The impact of technology on translation profession contemporary practices over the last two decades has been far-reaching and expanding [6] because of the development of new forms of MT and the steady popularization of Computer Aided Translation (CAT) tools and skillsets, including translation memory, terminology management tools, corpora, postediting and others [7]. The constant advancement in MT types, software, and performance since the beginning of the new millennium has created an urgent need among academic and non-academic communities of translators to build capacities in state-of-the-art digital literacies [8,9].

Throughout its evolution, automatic translation has witnessed a steady transformation that led to the emergence three distinct types of MT. The first type is Rule-Based Machine Translation (RBMT) [10], which represents a traditional automatic translation system that functions by retrieving linguistic patterns and transfer rules (morphological, syntactic, and semantic) from source language and target language monolingual or multilingual dictionaries to use them in a translation task. RBMT has its limitations, as it is not scalable, and it requires a great deal of time and effort to write the rules manually and update its

components. This explains why RBMT does not always succeed in translating lexical units with an implied semantic content or word-combination restrictions like collocations and idioms [11]. The second MT type is corpus-based MT [12,13] with its two forms: statistical MT (SMT), which prevailed before the emergence of neural MT, and example-based MT (EBMT), both of which are more efficient than RBMT, as they have an improved performance in cross-linguistic matching and do not require a high level of human involvement to be trained automatically.

The third form of automatic translation is neural MT (NMT), which employs artificial intelligence (AI) [14] and is, therefore, scalable, flexible and highly efficient in terms of speed, time and effort [15]. Artificial Intelligence (AI) is the ability of computer systems to perform tasks that are typically undertaken by human beings. Such tasks involve the development of deep learning models and problem-solving processes [16]. Neural machine translation uses artificial intelligence to learn languages and continue to improve that knowledge using neural networks. Based on statistical translation methods and the leveraging of extensive data and algorithms, NMT generates and develops neural networks that facilitate interaction between computers and humans, using natural language and, thus, empowering machines to understand and process human language.

While all machine translation systems implement the statistical approach of algorithms known as Statistical Neural Translation (SNT), earlier MT models such as RBMT and SMT translate limited linguistic structures, while NMT systems have revolutionized electronic translation tasks and processes by utilizing deep neural networks that are capable of translating more accurately, efficiently and fluently than previous MT models, as they incorporate cutting-edge neural networks and deep learning methodologies to augment their understanding of context and language structure [17].

Neural MT (NMT) systems such as Google Translate, Microsoft and Systran are "currently dominating the paradigms of machine translation" [11], (p. 595), since they are more reliable than earlier generations of MT, and their output can be adjusted by benefiting from human intervention. This intervention is referred to as Machine Translation Post Editing (MTPE). The progressive advancement in AI-driven NMT has made postediting a viable alternative to improve the quality and productivity of translation [18], as it enabled these engines to produce highly accurate content with certain restrictions relevant to text type, language pairs, style, and other factors.

### 1.2. Emerging Digital Literacies

Despite the steady development in MT software and tools, automatic translation systems have not reached full independence in producing quality output that qualifies them to replace human translators. Literature on the topic signals a shift in the role of human translators from being fully in charge of the translation process to becoming posteditors who follow up on the translation process before, during and after the application of automatic translation. Accordingly, machine translation output is not an accomplished product without human intervention; hence the importance of understanding the mechanisms of this involvement (MTPE) and investigating its advantages and drawbacks [19]. Ref. [20] highlighted the necessity to research MTPE, since MT output "is rarely published without some kind of post-editing" (p. 225).

Ref. [21] accentuated the need to reconfigure 21st-century translator competencies. MTPE is a digital literacy that still awaits due attention in the academic and professional practice of English/Arabic translation, considering the growing demand for translation and the increased use of MT among translators and translation students. While some research studies have underscored the popularity and recent surge in MTPE practice among users and translation service providers [22–24], there is humble recognition of postediting by translation communities, academic programs, and research in the English/Arabic language pair [25–28].

This systematic review is significant, as it highlights a poorly researched area in English/Arabic MT and a hotspot that requires urgent attention from translation pedagogues, practitioners and researchers, especially after the unprecedented growth in translation

demand and the noticeable popularity of MT use among language and translation learners worldwide. Also, the impact of the COVID pandemic on various levels of education has created a need to reconsider conventional teaching practices and introduce novel ones that respond to the requirements of digital literacy [29–33].

## 2. Conceptual Framework

### 2.1. MTPE: Affordances and Impediments

When MT started to gain popularity in the Arab world at the beginning of the new millennium, there was widespread uncertainty among translator communities about the quality of MT output and the efficiency of postediting the produced content in terms of time and effort. Predominant translator attitudes towards postediting MT output were rather poor and discouraging, as they deemed it to be more efficient to restart the translation process from scratch than postedit poorly translated content [19]. MTPE is an interaction between a human translator (HT) and a translation machine while editing MT output [34]. Ref. [19] defined postediting as a process undertaken by "a human being (normally a translator) comparing a source text with the machine translation and making changes to it to make it acceptable for its intended purpose" (p. 1). Ref. [35] remarked that postediting is "a bilingual language processing task . . . undertaken by experienced professional translators" (p. 106) to identify and correct errors or fix stylistic issues in a text translated by translation software.

Ref. [36] observed that postediting is the process of examining and improving the quality of a machine translation product in terms of correctness, accuracy, clarity, readability, style and other criteria identified by a translation brief. MTPE embodies an exemplary collaborative model between artificial intelligence and human translators, leading to a fundamental transformation in the practice of translation, considering the improved quality and reliability of MT performance and availability of free translation software [22]. The motives behind using MTPE are varied and include productivity gains (speed), developing a gist understanding of a source text, producing an improved quality of MT output in texts with high retrieval from translation memories such as technical or legal texts, reducing typing, and evaluating the quality of MT output [24].

According to [11], an objective evaluation of MT product quality "will require both automated and human metrics . . . because human evaluation and error annotation are extremely relevant when measuring MT quality—they are both processes that must be carried out by evaluators trained in the field of translation" (p. 594). Ref. [37] maintained that MTPE improves the productivity and quality of automatic translation processes, and [38] remarked that while users' perceptions about the productivity of MTPE refer to the consumption of extra effort and time, the actual postediting time needed for publishable translation quality is less than the time and effort spent on manually translated tasks. On the other hand, [39] signalled improvements in the quality, but not productivity, of MT output.

There are inconsistencies in the literature findings on the quality and productivity of MT output. These inconsistencies are related to direct and indirect factors regarding the potential of translation software, clarity of translation brief, similarity between involved language pairs, difficulties associated with text types and genres as well as translators' competence and experience in translation and postediting. MTPE is not merely a process of rectifying errors related to the wrong use of lexical items and patterns, grammar, punctuation, or the like. In fact, it is a challenging task which requires training and practice due to impediments related to the need to understand the workflow of complex systems, text segmentation efforts, as well as text formatting requirements. Ref. [18] remarked that "MTPE is a complex cognitive process which is closely associated with high-order thinking skills and self-regulatory strategies" (p. 341) that require critical thinking and emperical practice. Ref. [34] remarked that while there has been a growing demand for practicing MTPE, the popularity of postediting amongst the community of practicing translators is rather low for reasons to do with the poor quality of MT postedited product, compared to the product of human translation. The author viewed college learners as potential posteditors who should be trained in MTPE skills to improve the quantity and quality of

MT output. One of the major concerns about MT product quality is the social impact of uninformed MT use in certain specialized text types [40].

The concept of quality associated with MT application has gradually acquired the moderate perspective of being adequately good for a certain purpose. Also, quality is determined by the type and level of postediting required. According to [41], two types of postediting determine MT output quality: full postediting, leading to publishable quality, and rapid postediting, aimed at correcting errors for accuracy without refinement in style or fluency. Ref. [20] researched the role of ergonomic factors involved in professional translators' tendency to use MT. The authors defined ergonomics as the study of interaction between human beings and working-environment components, like computers, throughout their practice. The study concluded that most participants reported the beneficial use of MTPE in their daily translation tasks and that the views on adopting or not adopting MT are relevant to human factors such as translators' limitations, capacities and needs rather than the quality or productivity of MT output.

Researchers on MTPE view the prevalent reluctance and scepticism about postediting as biased [20] and prejudiced [34]. According to [34], one of the misconceptions that lead translators to develop sceptical attitudes about the nature and quality of MTPE in Japan is pertinent to the perceived assumption that it requires lower translation skills than human translation and may, eventually, result in undermining translators' skills over time. Also, translators who abstain from using MT report issues related to low-retrieval texts such as speeches, literary texts, press releases, etc., job security issues, as well as concerns about the negative impact of MT on translators' abilities.

One conventional argument against MT efficiency is the recurrence of the same errors produced by translation software, making translation posteditors feel irritated. However, the frequency of MT mistranslations became a resource for improving MT performance. Recently, MTPE evolved to Automatic Post Editing (APE); Ref. [42] reviewed research on developing APE from databases that have human postedited content. APE refers to the automatic process of improving MT output by using high-quality translation models postedited by human translators. For APE, MT systems require the availability of three sets of data by each textual segment: source content, MT-generated target content, and another version of target content corrected by HTs. The working mechanism of APE assumes that the third element in the data sets should not be a raw human translation produced without the interference of MT, because the objective behind using MT data patterns postedited by humans is to use them for "learning editing patterns for MT output" (p. 103).

Another form of postediting is an avantgarde model referred to as "online adaptation on NMT systems to interactive user post-edits" (321). Ref. [43] conducted a pioneering empirical study on the role of online human postediting in improving the performance of NMT. This postediting model occurs during real-time interaction between the human posteditor and the automatic translation system whereby translators apply their postediting Input to MT output online. The study was implemented on a sentence per sentence MT from English into German, and the results showed "a significant reduction in post-editing effort" (p. 310). Similar studies were previously conducted on the efficacy of generating online human postedits to improve NMT output on the level of the phrase.

## 2.2. MTPE in Translator Education and Training

Since MTPE has gained popularity in the translation industry, it requires more attention in the academic field [18], particularly following the COVID-19-triggered transformation to online teaching which has affected educational practices in academic institutions worldwide [3,31,32]. While there was a growing demand for using MTPE, translators received little training in the skills of MTPE [44]. Training translation learners to postedit texts translated by machine software should take place as part of a MT course rather than separately. Due to scepticism about MT potential to replace human translators, integrating MTPE with translation curricula emerged with some delay in 2009 [36], and until now many

universities that offer translation programs, particularly in Arab countries, undervalue the provision of postediting training at advanced levels.

For the last three decades, there have been calls by translation academics and researchers to integrate MTPE skills with translator education programs [21,44,45], highlighting the importance of pre-editing and postediting the input to boost the efficiency of Human-Aided Machine Translation (HAMT). Furthermore, posteditors need to acquire the programming skill of writing macros as they develop an experience in frequent MT errors. Mastering macros paves the way for developing APE software. Trainee posteditors should also develop their skills in text linguistics to improve MT output and become macro programmers. Ref. [19] believed that translators need training in MTPE, and translator training programs need to instil in posteditors a diverse set of skills, including linguistic and cross-linguistic skills, professional translator skills in using translation strategies and making relevant decisions, as well as technical skills focused on "advanced knowledge of computer software functions" (p. 17).

While conventional arguments against MT concentrated on the potential of machines competing with human beings or replacing them, recent trends in MT workflow popularize the role of translators as future posteditors. Ref. [44] observed that nurturing postediting skills in training translators enhances their employability and makes them comfortable with MT and more empowered to harness AI for their own benefit. Furthermore, practicing postediting skills facilitates learners' understanding of its affordances. Therefore, it is important to teach postediting in translation courses. Ref. [44] outlined the skillsets that should be included in a postediting course. Posteditors should develop knowledge and appreciation of MT technologies, their fundamentals, advantages and limitations. Also, postediting skills include terminology management competencies in knowing how to store and retrieve terms. Another skill to be covered by a postediting training package is mastering the use of controlled language (language modified by pre-editing), as this improves MT output quality.

Ref. [45] underscored the importance of improving translation students' understanding of MT errors by integrating MT output error analysis with translation teaching and training. The study provided analysis of texts postedited by translation students and showed that students' postediting efforts were subconscious and superficial and need to be enhanced with educational guidelines to take their postediting exercise to a deeper level that allows for phraseological and stylistic improvements. Ref. [46] pointed out that MTPE is an emerging translator competency which still lacks clear guidelines that inform trainees in translation programs and practitioners in professional environments. The study investigated the understanding of MA trainee translators of available postediting guidelines and indicated the necessity to address gaps in translator trainees' MTPE competencies.

Ref. [47] remarked that educating new translator generations does not address the transformation in the workflow of modern translators adequately. Translation education and training programs need to bridge the existing gaps between graduates' competencies and labour market demands. The author provided a proposal for a translation course that can respond to translation students' need for "technological literacy" (p. 28), which refers to translators' mastering of various technological translation tools, including CAT tools, TM management tools, MT tools and processes, etc.

Ref. [36] investigated the value of introducing postediting as a translation competency in undergraduate translation curricula in China. The purpose was to bridge a gap between translator education and training in academic programs and practice in the translation industry, considering the rising demand for MT accompanied by a parallel need for the provision of human postediting. According to the authors, the application of postediting can be facilitated by enhancing the quality of MT output. This includes several steps, starting with determining the translation brief to decide on the type of postediting (light or full postediting) and conducting pre-editing (checking on language, format, simplifying syntactic structures and explicating vague content, as well as scrutinizing terminology).

MTPE continues to gain popularity in MT research, and this should be reflected on the academic level in translator training and translation pedagogies that aim to bridge the gaps between translation programs' outcomes and translation profession requirements. Ref. [48] provided a rich review of MT research from the angle of language and translation studies to identify trending issues and hotspot research areas. The study concluded that research in MT was dominated by the themes of AI-empowered NMT and the integration of human postediting to improve MT product, both of which emerged as prominent research topics that can enlighten future studies on MT to improve automatic translation processes and performance.

This study marks a departure from relevant reviews on the topic in that it focuses on research contributions to MTPE in Arabic/English MT. The study provides a quantitatively informed synthesis and analysis of the relevant literature by extracting and interpreting data from published research on the topic to provide evidence and critical interpretation of dominant research trends and inform future research and practices in the field of Arabic/English MT. The study attempts to answer the following questions:

(a)  How has research on English/Arabic MT evolved since the beginning of the twenty-first century?
(b)  What is the status of English/Arabic MT in terms of focus areas, gaps and emerging trends?
(c)  What are the most prominent research approaches and methods in English/Arabic MT, and how reflective are they of MT training on the academic level?

## 3. Methods and Instruments

### 3.1. Research Methodology

This systematic review was conducted using the Preferred Reporting Items for Systematic Review and Meta-Analysis Protocol (PRISMA-P) in data identification, screening, synthesis and analysis. The research methods aimed to combine and quantify collected data to synthesise common metrics, provide an unbiased analysis of main trends and differences in the results of reviewed studies, and summarize the developed empirical knowledge. This research method is common in different disciplines including education. However, because PRISMA-P does not inform the processes of data extraction and synthesis methods in translation studies, the research methods were adapted accordingly [2,49].

### 3.2. Literature Identification and Screening

A literature search was conducted using Clarivate Endnote Google Scholar button to retrieve sources based on relevance to research topic and questions, language pairs and publication date. The purpose behind choosing Google Scholar is that it indexes different types of scholarly literature, including articles, theses, books, and conference papers, which allows for an unbiased approach to data identification and screening. The sources were identified and screened over the period October–November 2023, covering studies published during the timespan between 2000 and 2023 and adopting clear inclusion and exclusion criteria to ensure the validity of reviewed studies in terms of alignment with the study objectives. The inclusion criteria comprised studies' relevance to the research topic, research language (studies written in English), journal indexing, peer-review or academic research status, as well as relevance to the MT language pair (English/Arabic).

The researchers keyed in "Machine Translation postediting OR post editing English Arabic" anywhere in the article and retrieved 370 results before screening. Provisionally, the raw results were screened in terms of titles and abstracts and sometimes full manuscript by applying the inclusion and exclusion criteria. The first round of screening yielded 75 results. Studies which did not fulfil the inclusion criteria were excluded. Duplicates were also excluded from the search results. For example, the researchers excluded studies that did not approach MT from the perspective of translation studies. Also, studies that discussed the topic in relevance to language combinations other than English/Arabic were excluded.

In the second round of screening, the results were filtered in terms of full manuscript content and academic publishing status, excluding studies not published in peer-reviewed and indexed journals. The number of studies considered for the review was narrowed down

from 75 to 60 studies. Throughout the screening phase, the authors referenced the list of collected studies according to APA 7th edition after correcting errors in authors' names, studies' titles, publishers' details, etc. Since the tokens included academic studies, published books, book chapters, peer-reviewed published papers and conference papers, no quality assessment criteria were adopted in screening the studies regarding the implemented research methods used or the validity of results, as the purpose of systematic literature review using PRISMA-P is to benefit from an unbiased approach to data collection, synthesis and analysis.

*3.3. Data Extraction, Synthesis and Analysis*

After filtering the collected materials, the researchers identified various data types to be extracted from the corpus of literature for synthesis, tabulation, and analysis. The full content of all sixty studies was accessed for accurate data extraction. The extracted data included publication date (citations to facilitate data synthesis and analysis and discussion of results), research type or journal ranking, language-pair direction (English–Arabic, Arabic–English, or both), research focus and research methods, MT software(s), as well as text type(s) or textual components. The purpose behind classifying and extracting these data is to produce quantitative metrics that can be synthesized and analysed for the purpose of TS research. These metrics are helpful in tracking the development of the research topic and highlighting research trends, issues, and gaps. They provide TS researchers, pedagogues, and practitioners with a clear understanding of the latest practices in the field and future directions for translation research and education development. Table 1 below provides a sample of data extraction by different metrics:

**Table 1.** Sample of representational data collection.

| Citation | Approach and Framework | Research Methods | MT Systems | Research Focus | Research Type | Lang. Pair | Text Type |
|---|---|---|---|---|---|---|---|
| [50] | Evaluative: User model MT effectiveness | Mixed | Four systems:<br>1. Translate Dict.<br>2. Yandex<br>3. Mem-Source<br>4. Reverso | Comparing HT with MT. Results highlighted MT inadequacy and need for pre-/postediting | Journal of Language and Linguistic Studies (Scopus) | ENG-ARA | Literary texts |
| [51] | Evaluative: Survey of previous studies by:<br>1. Reliability<br>2. Fidelity<br>3. Terminology<br>4. Syntax | Mixed Comparative | Seven systems:<br>1. GT<br>2. Ajeeb<br>3. Professional Translator<br>4. 1–800 translate<br>5. World lingo<br>5. Tran Sphere<br>6. An-Nakel | Advancement in productivity of Arabic MT GT compared to other systems (2008–2013) | Educational Research and Reviews Science Direct IF | ARA-ENG | 1. Technical<br>2. legal<br>3. literary<br>4. journalistic<br>5. economic |
| [52] | Evaluative: MT linguistic limitations | Qualitative | Three systems<br>1. Systran<br>2. GT<br>3. MB | Identifying linguistic and morphological errors:<br>1. Subject-verb agreement<br>2. Adjectival-noun agreement<br>3. Pronoun-antecedent agreement | Procedia Computer Science, (Elsevier-IF) | ENG-ARA | Gender bound constructs in technical texts |
| [53] | Evaluative: Investigating MT accuracy | Descriptive | GT | Complementarity between MT and HT: MTPE to improve translation accuracy | Journal of Reproducible research (Indexed and peer reviewed) | Both | 1. Technical texts<br>2. literary texts |
| [54] | Evaluative: Failure of fully automated MT (without pre-/post editing) in:<br>accuracy<br>correctness<br>acceptability | Mixed: error analysis | Four systems:<br>1. Tarjim<br>2. Ajeeb<br>3. Almisbar<br>4. Al Wafi | MT limitations in dealing with contextuality, culture-bound expressions, lexical and structural ambiguity, and idiomatic expressions. | International Journal of Arabic-English Studies (Scopus as of 2016) | ENG-ARA | 1. Scientific<br>2. Specialize<br>3. Technical<br>4. Legal<br>5. General |
| [55] | Role of translation evaluation in improving MT systems:<br>adequacy<br>fluency | Mixed: error analysis | Three systems:<br>1. Google Translate<br>2. Microsoft Translator<br>3. Sakhr | Evaluation criteria: Adequacy and fluency GT has highest score followed by MT and Sakhr | Unpublished Doctoral Dissertation University of Western Australia | Both | 1. Literature<br>2. UN docs<br>3. Arab League Docs |
| [56] | Producer model: Building a sentence-aligned, error-tagged undergraduate learner translator corpus | Mixed | None | Learner translator corpora as a resource for translation Pedagogues and researchers in MT experimentation to fill in a gap in ENG-ARA translation resources | Language Resources and Evaluation (Springer) IF | ENG-ARA | 1. Political<br>2. Medical<br>3. social |

## 4. Results and Discussion

### 4.1. Literature Taxonomy by Study Types, Language Pair, MT Engine and Text Types

This section provides the results sorted by study types, MT engines, language-pair direction as well as text types investigated across all literature throughout the period 2000–2023. The purpose behind classifying data by these metrics is to identify relevant research gaps and trendy aspects that dominated English/Arabic MT research for the last twenty-three years. The following tables provide the results.

Table 2 below provides a frequency count of the literature by study type or publishing ranking. The aim behind sorting the literature by publication type and ranking is to assess the quality of existing research on Arabic MT, evaluate the overall research output on the topic and investigate its status in terms of scholarly recognition, excellence and advancement. This evaluation can inform strategic decisions and resource allocation and can be valuable for identifying trends, areas of expertise and collaborative opportunities. Furthermore, classifying the literature by different study types or ranking, such as journal articles, conference papers, books, and reviews, helps researchers use these classifications to locate information relevant to their needs.

**Table 2.** Literature taxonomy by study type/ranking.

| Study Type/Ranking | Number | Percentage |
|---|---|---|
| IF | 12 | 20% |
| Doctorate/MSc. | 15 | 25% |
| Scopus paper | 10 | 16.6% |
| Book/book chapter | 2 | 3.3% |
| Conference paper | 6 | 10% |
| Other | 15 | 25% |
| Total | 60 | 100% |

Table 2 shows that the highest percentage of study types was that of doctorate or master's degree dissertation and journal papers published in indexed venues other than Scopus or Impact Factor journals, 25% each. This result suggests that there is visible research interest in this area and that it has reached a reasonable level of maturity. This result shows that there is academic recognition of the significance of this topic and indicates that educational programs and academic institutions support research in this field. Studies published in IF journals reached a percentage of 20%, ranking next to doctorates or master's degree theses, and followed by studies published in Scopus-indexed journals (16.6%). This result suggests that research on English/Arabic MT is associated with great visibility and impact. The increased visibility of studies reflects an expansion in the topic's dissemination and access to funding opportunities. It is also indicative of the topic's influence on relevant policies and practices in academic institutions.

The lowest percentage for study type is that of published books (3.3%), which is a very low percentage compared to other metrics. This result suggests that the topic falls within an emerging field. But putting this in the context of the covered duration (2000–2023) and the number of published papers suggests a slow progression of the topic. The fact that the number of published papers exceeds that of published books indicates the researchers' priority to publish their studies in peer-reviewed journals to share the latest research and achieve academic recognition quickly, considering the rapid changes affecting the topic and its practical nature.

While study type and publication ranking provide significant information about the scholarly status of a topic, they are not the sole indicators to be considered by researchers and institutions. Other factors, including the study context, intended audience, and impact within the field, are equally important. Additionally, relying solely on publication metrics for evaluating literature on a certain topic involves some bias; hence the importance of

diversifying data synthesis methods. These metrics will be further elaborated vis-à-vis the results on conceptual and historical evolution. Figure 1 provides a visual representation of study types and publication rankings.

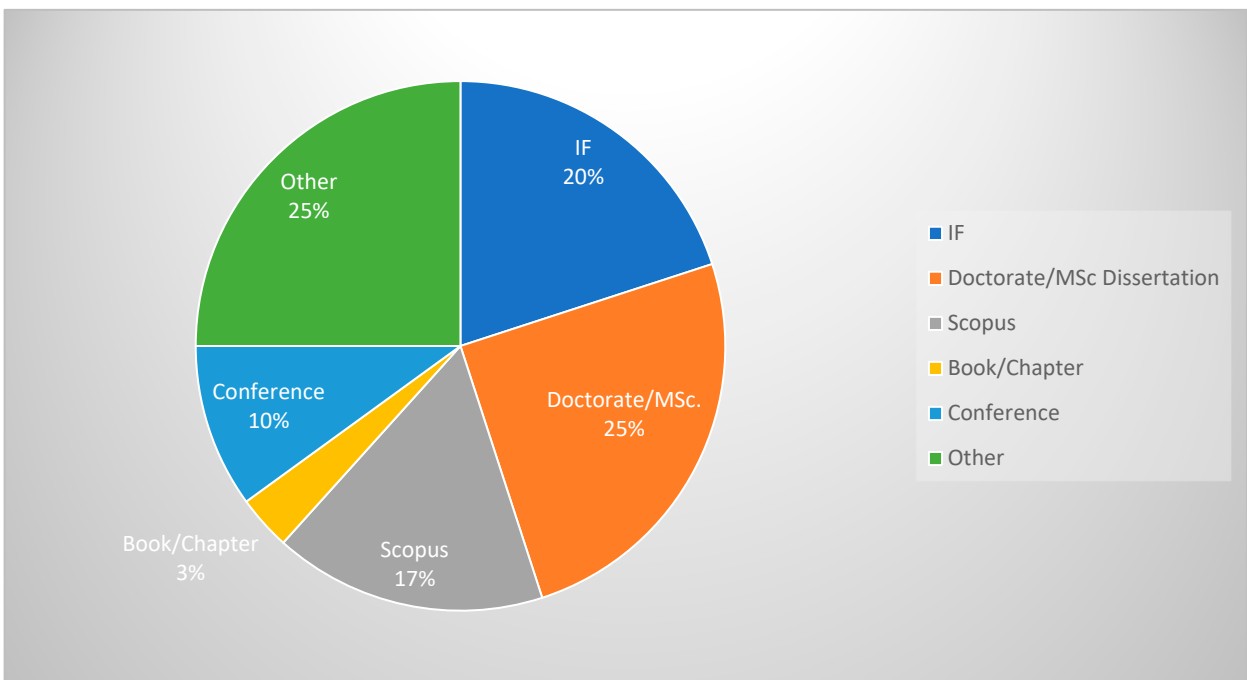

**Figure 1.** Study type/ranking frequency.

Table 3 below shows the distribution of language pairs across different studies. Of 60 studies, 48 researched different aspects of English/Arabic MT in relation to one direction of these two languages at least. Studies that did not address English/Arabic MT topics in terms of their application in a language-pair direction were either perceptual, historical reviews or technically focused. Section 2 of the results and discussion will trace the topic's development in terms of the studies' approach and focus. The preliminary results show that most studies (39.5%) focused on investigating MT output in English–Arabic translation compared to 29.16% that focused on Arabic–English translation and (31.25%) that focused on both directions of the language pair.

**Table 3.** Frequency and percentage of language-pair direction.

| Language-Pair Direction | Frequency | Percentage |
|---|---|---|
| English–Arabic | 19 | 39.5% |
| Arabic–English | 14 | 29.16% |
| Bi-directional | 15 | 31.25% |
| Total | 48 | 100% |

This result is significant, as it reflects the status of English/Arabic translation in general. Although the difference in the number of studies that researched Arabic–English MT compared to studies that researched English–Arabic translation is not highly significant, the results signal the need for conducting more studies that investigate MT in relation to Arabic–English. While some studies concluded that error frequency in Arabic–English translation is less than that in English–Arabic translation [57,58] stressed the need for postediting to improve MT adequacy and fluency in Arabic–English translation. The following chart provides a visual representation of results in terms of language-pair direction.

Table 4 below provides a statistical representation of Arabic MT engine occurrence in the literature. It is worth noting that not all studies researched the potential of available MT software. There are certain studies that researched the affordances of other translation resources such as online dictionaries like Almaany [26] and Translate Dictionary [50]. Ref. [51] investigated the performance of MT systems that seem to have been discontinued or that experienced a change in their status, such as 1–800-translate, which shifted to telephone interpreting services and Ajeeb (introduced in the study as Systran free translation) which can no longer be traced on the web. Al-Kafi seems to have been discontinued [59]. These were excluded from the frequency count, as this review is focusing on MT software that is still in use.

**Table 4.** Frequency, percentage and subscription of Arabic MT software.

| | MT Engine | Frequency | Percentage | Subscription |
|---|---|---|---|---|
| 1. | Google Translate (GT) | 28 | 29% | Free |
| 2. | Systran | 11 | 11.30% | Commercial |
| 3. | Microsoft Bing (MB) | 8 | 8.24% | Optional |
| 4. | Reverso | 8 | 8.24% | Free |
| 5. | Tarjim (Sakhr) | 6 | 6.18% | Free |
| 6. | Yandex | 5 | 5.15% | Commercial |
| 7. | Babylon | 4 | 4.12% | Commercial |
| 8. | AppTek | 4 | 4.12% | Commercial |
| 9. | An-Nakel | 3 | 3.09% | Optional |
| 10. | Al-Wafi | 3 | 3.09% | Commercial |
| 11. | Memsource | 2 | 2.06% | Commercial |
| 12. | Tran Sphere | 2 | 2.06% | Commercial |
| 13. | Ajeeb (Sakhr) | 2 | 2.06% | Commercial |
| 14. | Almisbar | 2 | 2.06% | Free |
| 15. | Al-Mutarjim Al-Arabey | 2 | 2.06% | Optional |
| 16. | Professional Translation | 1 | 1.03% | Commercial |
| 17. | WorldLingo | 1 | 1.03% | Optional |
| 18. | Weinder | 1 | 1.03% | Commercial |
| 19. | Lilt | 1 | 1.03% | Commercial |
| 20. | Mutarjim Net | 1 | 1.03% | Commercial |
| 21. | Ginger | 1 | 1.03% | Free |
| 22. | Collins | 1 | 1.03% | Commercial |
| 23. | SDL Trados | 1 | 1.03% | Commercial |

Table 4 shows that the literature investigated at least 23 different types of Arabic MT systems. The highest frequency among researched Arabic MT engines is that of GT, followed by Systran, MB, Reverso, and Tarjim, respectively. Also, most studies concluded that GT had the best performance in terms of different criteria, including adequacy, frequency, accuracy, time-efficiency, productivity, context sensitivity, suitability, adaptability, flexibility, terminology, probability, error frequency as well as control and learnability [60–64]. This result is in line with the findings of [11] that GT, MB and Systran are dominating MT paradigms.

However, the literature has also shown variation and discrepancies in evaluating Arabic MT engines. For instance, some studies show that GT was the least accurate in terms of intelligibility [65] or in dealing with legal discourse. Ref. [60] concluded that while both GT and MB achieve a percentage of accuracy, exceeding 90% in translating journalistic texts, the former has a slightly better performance than the latter in translating

collocations. Ref. [65] reported that MB was the best engine, while [26] concluded that Reverso Context has a similar performance to GT. Ref. [66] concluded that GT has a slightly higher evaluation than Reverso. Ref. [25] reported that Systran, Reverso, and Yandex achieved similar results in terms of accuracy. Other studies focused on different types of meanings. Ref. [67] concluded that while GT is better in translating semantic content, Systran and MB were more effective in translating communicative meaning.

This result shows that studies on evaluating Arabic MT engine performance have overlooked evaluation criteria related to crucial considerations like the societal implications of applying MT by incognizant users. This result consolidates the findings of the critical review by [40] on the gap in addressing social issues in research on MT of legal and health care texts. The fact that the literature on Arabic MT focuses on evaluating the machine performance without taking into consideration the binary relation between machine translators and human translators resulted in partial treatment of the topic without addressing translators' competencies essential for effective postediting. This result highlights the need for conducting more evaluative, comparative studies on the performance of different Arabic MT engines with particular focus on MT quality parameters that require high-order competencies and critical thinking in processing different text types and textual features. Figure 2 provides a visual representation of Arabic MT systems investigated in the literature.

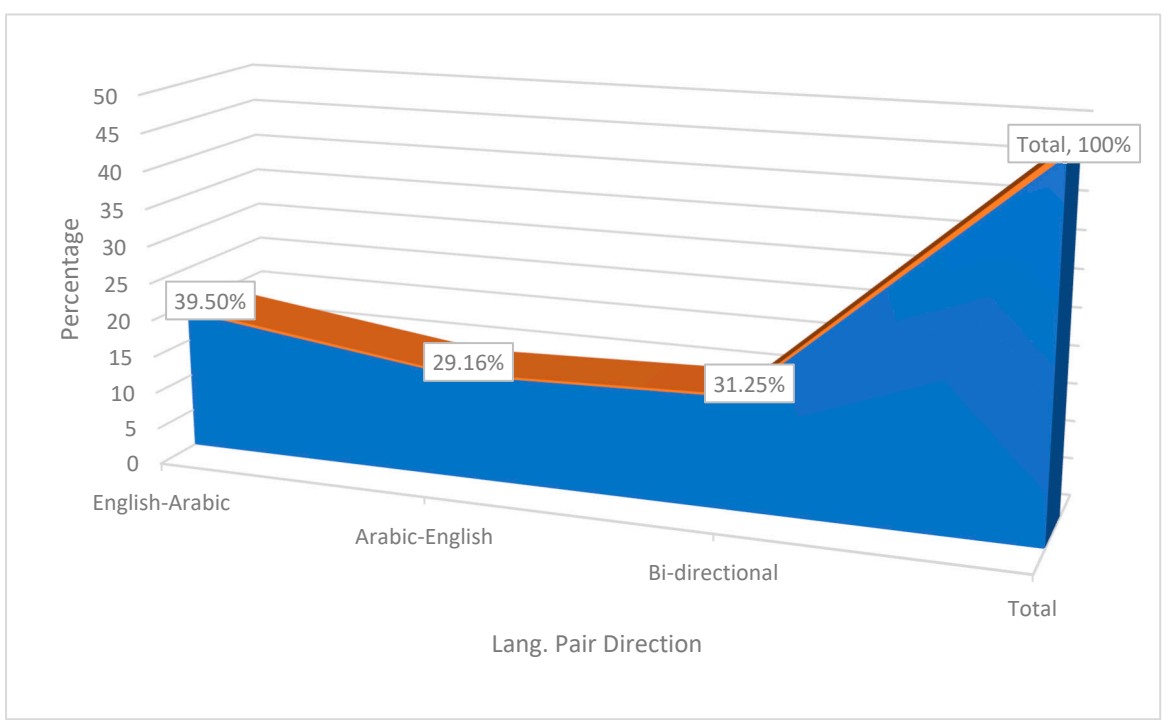

**Figure 2.** Frequency of language-pair direction.

Figure 3 shows that the most prominent Arabic MT systems are those with free subscription, such as GT, MB and Reverso. Nevertheless, ranking Systran next to GT in terms of frequency shows that commercial MT engines are also considered for the purpose of research despite being less popular than free software. Ref. [68] indicated that al-Mutarjim Al-Arabey used to be purchased, while recent updates show that this system is available for free. Memsource provides a free trial for a limited period, while others provide optional subscription, such as MB, An-Nakel and Al-Mutarjim Al-Arabey. Ref. [69] provided a historical overview of Arabic MT software without listing relevant subscription details. The results highlight the significance of reviewing available Arabic MT systems in terms of different parameters like comparative evaluation and subscription details.

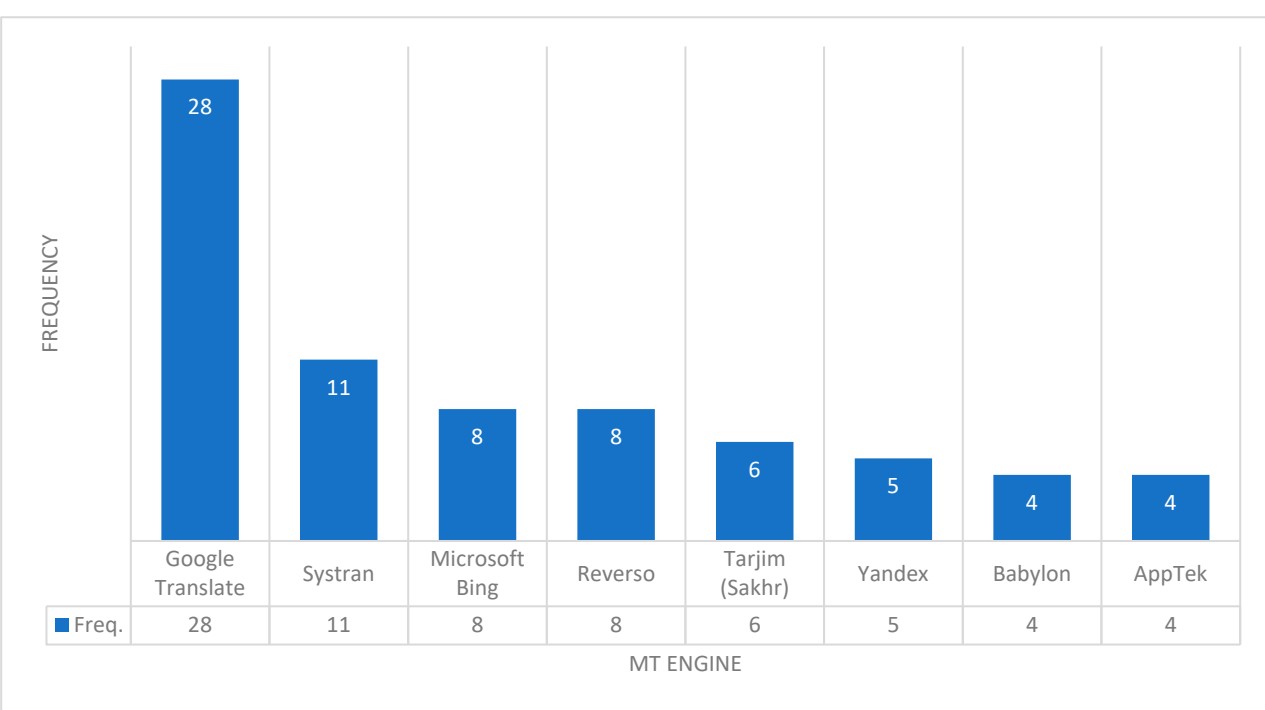

**Figure 3.** MT engine frequency.

Table 5 below provides data frequency by text types. The table shows variation in Arabic MT research in terms of text types. Different studies investigated a variety of text types including literary, scientific, technical, cultural, journalistic and social media. The data were clustered and sorted in terms of relevance and proximity. For example, while some studies researched the efficacy of Arabic MT systems in dealing with certain literary genres like fiction [66], others focused on a specific feature of different text types such as idiomatic expressions in literary texts [70] and collocations in journalistic texts [60]. These were clustered together in view of proximity in meaning type (connotative and expressive). Similarly, technical and scientific texts were also clustered together in view of proximity in their function (informative).

**Table 5.** Data frequency by text type.

| Text Type | Frequency | % |
|---|---|---|
| Literary texts and features | 5 | 8.6% |
| Scientific and technical | 12 | 20.6% |
| Cultural and rhetorical | 5 | 8.6% |
| Legal and UN documents | 14 | 24.1% |
| Journalistic and editorial | 4 | 6.8% |
| Social media and dialectal | 2 | 3.4% |
| Online books and movies' reviews | 3 | 5.1% |
| General | 5 | 8.6% |
| Total | 58 | 100% |

Figure 4 below shows that the highest percentage of data frequency in terms of text types is that of legal and UN discourse (24.10%), followed by scientific and technical texts, which reached a percentage of 20.60%. Legal discourse and scientific texts share the attribute of being predominantly informative, which makes their processing by machine less challenging than other text types. Results show the need for more research on expressive

text types such as literary, cultural, and rhetorical texts, all of which reached the same percentage of general texts (8.60%). Results also show a lack of studies that researched Arabic MT effectiveness in translating journalistic texts (6.80%), dialectical and social media content (3.40%) and online reviews (5.10%).

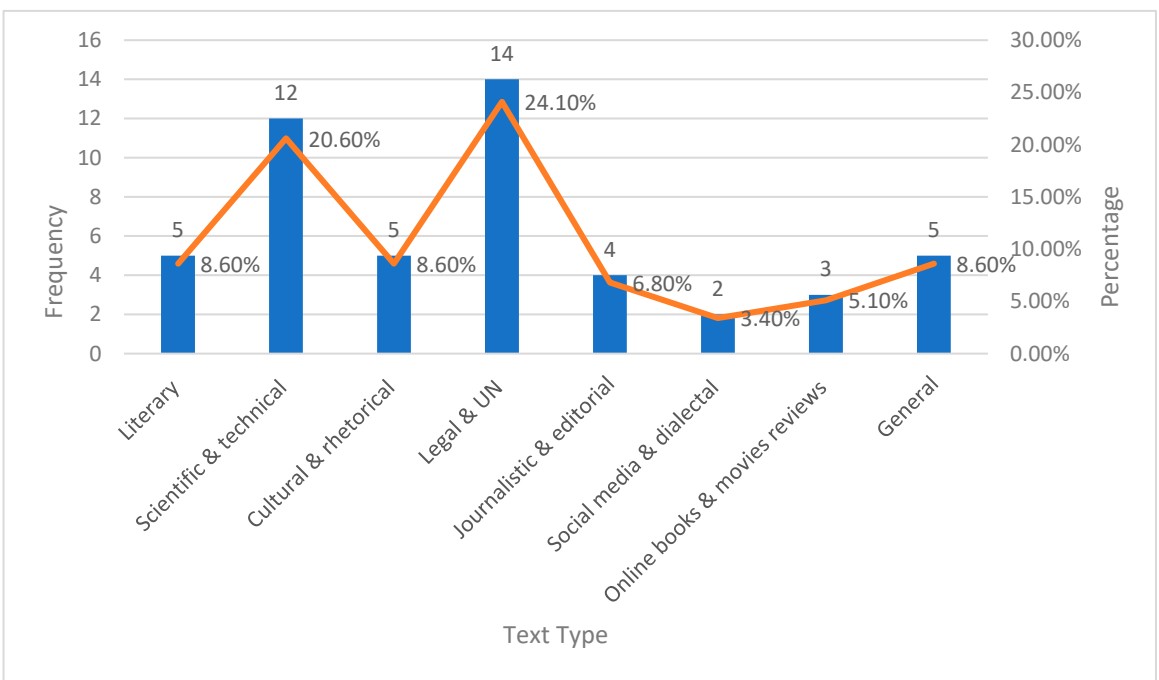

**Figure 4.** Data frequency by text type.

The following section provides the results and analysis in terms of chronological and conceptual progression in Arabic MT research. The results in this section will show whether there has been development in research on Arabic MT in terms of the number of studies, their investigation of MTPE as well as the adopted research methods. The purpose behind this focus is to locate the status of Arabic MT research on the international map of similar research and identify conspicuous conceptual frameworks and emerging ones.

### 4.2. Literature Chronological and Conceptual Evolution

This section provides the results sorted by publication date and conceptual framework to highlight chronological and conceptual development in scholarly contributions to the topic and identify dominant trends and existing gaps. The publication timespans were divided into three periods: 2000–2010, 2010–2020 and 2020–2023, as each time span marks the emergence of a new trend in Arabic MT use and evolution. Table 6 below provides data collection results and taxonomy by literature publication dates and approach.

**Table 6.** Literature distribution by publication date and approach.

| Duration | Number | % | Approach and Framework | |
|---|---|---|---|---|
| 2000–before 2010 | 12 | 20% | - | Evaluative (user-oriented) |
| | | | - | Empirical/technical (producer-oriented) |
| 2010–before 2020 | 20 | 33.3% | - | Evaluative |
| | | | - | Perceptual |
| | | | - | Explorative |
| | | | - | Semi-pedagogic |
| | | | - | Reviews |
| | | | - | Empirical technical |

**Table 6.** *Cont.*

| Duration | Number | % | Approach and Framework | |
|---|---|---|---|---|
| 2020–end 2023 | 28 | 46.6% | - | Evaluative |
| | | | - | Semi-pedagogic |
| | | | - | Empirical pedagogic |
| | | | - | Perceptual |
| Total 2000–2023 | 60 | | 100% | |

In general, the results show a steady evolution in the number of publications on English/Arabic MT research, with most studies published after the COVID-19 outbreak in 2020. It is worth noting that the pandemic influenced educational and professional practices in translation vis-a-vis MT applications in view of the complete and abrupt shift to online education and in response to the increased demand of urgent translation work to disseminate information about the latest developments and precautions worldwide [71]. Figure 5 below shows that 46.6% of the literature published on the topic since the beginning of the new millennium emerged following 2020, which signals a transformation in scholarly interest in English/Arabic MT.

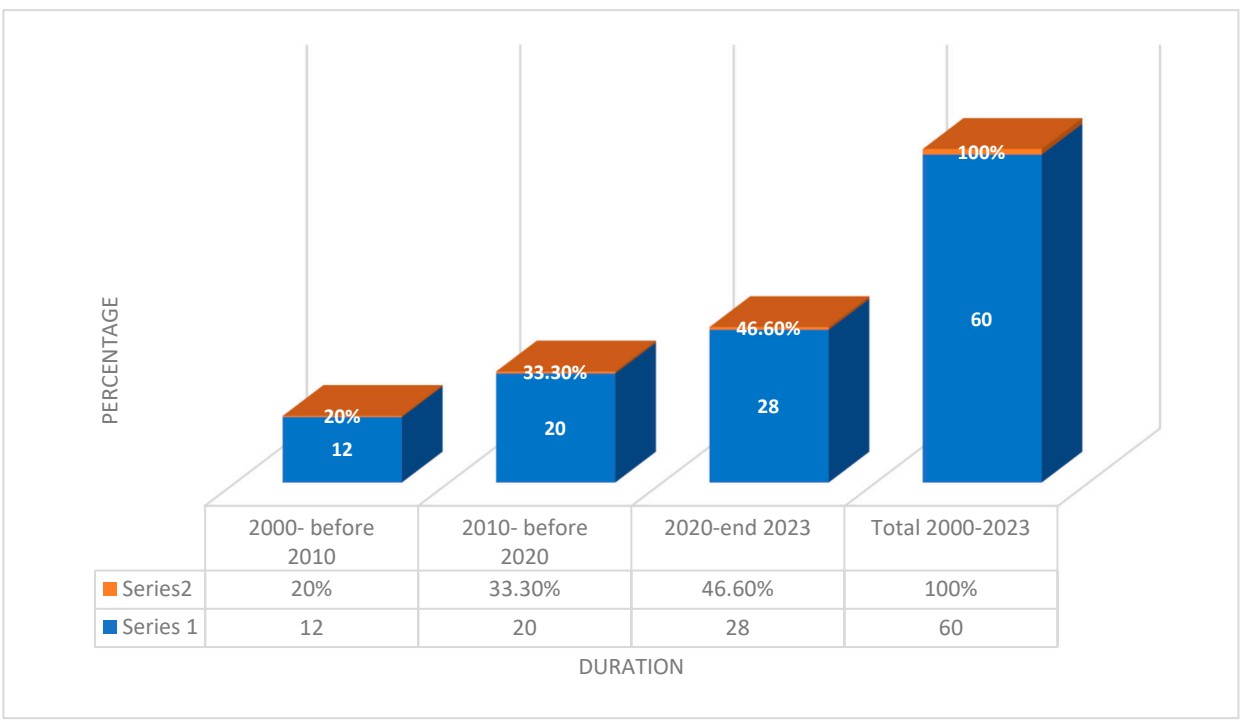

**Figure 5.** Literature distribution by publishing date.

4.2.1. Between 2000–2010

The number of studies that dealt with the topic before 2010 reached 12 studies (20%) that adopted either an evaluative model or a technically oriented experimental approach. Studies that adopted an evaluative approach aimed to test the accuracy, acceptability and readability of MT output in dealing with different types of content, including contextual, cross-cultural, idiomatic, or pragmatic language uses [54,59,72]. All studies that adopted an evaluative approach highlighted MT systems' insufficiency and limitations and the need for collaboration between MT systems and HTs by means of pre-/postediting. Evaluative studies focused on the types of errors produced by MT systems to outline the development of Arabic MT engines from a comparative, descriptive or chronological perspective [69] and suggest solutions to improve MT systems [68,73] and reduce human postediting efforts by improving MT processing of lexical properties and collocations [73] and expanding

parsing devices and syntactic analyzers [59]. This scholarly focus on translation errors, as opposed to deep translation issues, in evaluating MT performance, is justified during the early stages of Arabic MT evolution when MT engines were still not capable of producing adequate translations.

Most studies that adopted a technically oriented empirical approach were academic studies, either doctorates or MSc. research. These studies focused on developing different types of MT software. Ref. [74] concentrated on developing small-scale MT systems to deal with small messages and sublanguage domains. Ref. [75] focused on improving SMT of phrases in view of limited bilingual training data and the inability of MT to integrate syntactic and semantic information available on training data. Ref. [76] discussed supplementing RBMT systems with contextual and morphological information by using NLP information extraction to improve MT processing of named entities in Arabic texts. Ref. [77] addressed the lack of symmetry between English and Arabic by exploiting symbolic and statistical target-language resources (Heavy Hybrid Machine Translation system). Ref. [78] focused on developing a transfer-based MT system to deal with noun phrases in scientific and technical documents. Ref. [79] addressed the use of Universal Networking Language as an interlingua to generate natural language.

The point in common across studies that were conducted before 2010 is their pursuit to identify shortcomings in MT systems and improve their performance. This trend is characteristic of an early phase of MT use in English/Arabic translation. Although the development of Arabic MT systems was still in its early stages then, with sharply defined gaps between MT performance and HT performance, there was growing interest in integrating the automatic translation model with human intervention at an advanced phase of MT software development.

### 4.2.2. Between 2010–2020

The number of studies published during the second phase of Arabic MT evolution reached 20 (33.3%) studies that adopted an evaluative, experimental/technical, perceptual, semi-pedagogic or review approach. Studies that adopted an evaluative approach aimed to highlight limitations or outline advancement in Arabic MT systems. Research that pursued an evaluation of MT systems in terms of accuracy focused on the limitations of MT on the linguistic level, including syntax, morphology, word order, tense, aspect, etc., highlighting grammatical errors such as subject–verb agreement, adjectival–noun agreement or pronoun–antecedent agreement [52]. Ref. [80] evaluated MT accuracy in terms of linguistic obstacles that obstruct the production of accurate output in the case of polysemous words. Ref. [81] evaluated the effectiveness of free MT systems in terms of precision. Evaluative studies continued to highlight MT output shortcomings while marginally stressing the need for human intervention in the form of postediting [80]. Some evaluative studies surveyed and explored the progress achieved in certain MT engines compared to others. Nonetheless, the continual improvement in the performance of NMT engines was not paralleled by a shift in research focus on critical issues and challenges encountering MT users, such as ethical considerations [82], social implications [40] as well as cross-cultural communication [83].

The second phase of evolution in Arabic MT research was also marked by the emergence of review studies. Ref. [84] surveyed MT evolution for three generations, starting with the Direct Method and followed by the Transfer Approach and SMT, highlighting the success achieved in Arabic SMT Systems by incorporating linguistic knowledge. Ref. [51] provided an evaluative review of previous studies (2008–2013) on seven Arabic MT engines according to the functional criteria of reliability, fidelity, terminology, and syntax. The study highlighted the progress achieved in the productivity of GT compared to other systems. Ref. [85] provided a review of research on MT history, types and electronic processes to help researchers highlight the tools needed for further improvement. Ref. [86] surveyed studies on MT of Arabic dialects as an underdeveloped research area and explored the potential of developing an Arabic engine that translates dialects into Modern Standard Arabic. Reviews on the topic identified areas of progress and aspects that needed researchers' attention.

None of the reviews embraced a holistic approach to the literature with a view to compile, select, and synthesize available evidence for a clear definition and evaluation of trends and gaps in available research.

As for studies that adopted an empirical producer-oriented model focusing on the technical aspects of Arabic MT, they were predominantly doctorates [87–90] that addressed MT accuracy issues by improving existing MT systems. Ref. [87] investigated the affordances of crowdsourcing annotation in sentiment analysis with the purpose of editing MT output and distinguishing positive movie reviews from negative ones. Ref. [88] focused on improving SMT by incorporating linguistic components such as morphological splitting, syntactic reordering, and lexical contextualizing to remove opacity. Ref. [89] experimented with the endeavour of building large-scale phrase reordering models to increase MT productivity while preserving output accuracy. Ref. [90] focused on improving phrase-based SMT language modelling by proposing new methods and algorithms in transliteration mining, domain adaptation, and word-meaning disambiguation.

Producer-oriented studies were all developmental, aiming to introduce progress in the function of Arabic MT. Ref. [91] investigated the integration of postediting and interactive MT approaches exploring the possibility of postediting and improving raw MT output monolingually without source language knowledge. Ref. [92] investigated the use of the rule-based approach coupled with contextual and morphological information for named-entity recognition in Arabic texts to improve MT performance. Ref. [93] researched building a manually postedited MT corpus of Modern Standard Arabic to improve MT quality by error correction and develop automatic postediting systems for Arabic to accelerate human revision. This study overlaps with that of [42] on the potential of developing APE from databases that have human postedited content. Ref. [94] researched the development of a prototype English/Arabic MT engine to address MT accuracy issues.

The period between 2010–2020 also witnessed the emergence of perceptual or semi-pedagogic studies, signalling a shift in the status of English/Arabic MT. Studies described as semi-pedagogic incorporated a pedagogic framework dealing with educational principles and practices vis-à-vis incorporating MT with translator education without providing a clear description of instructional methods and learning facilitation strategies. Ref. [95] conducted a perceptual study that surveyed MT status in KSA. The study signalled a lack of interest in MT on the scholarly, professional, and academic levels. Refs. [96,97] conducted semi-pedagogic studies that explored the impact of MT and CAT tools on translation students' performance and future employability and the need to integrate technology into translation curricula and instruction. These studies are in line with [19,21,34,44,45,98,99] on the need to integrate MTPE with translation education programs; nonetheless, they did not adopt an empirical pedagogic approach to postediting, which signals an underrepresentation of this emerging skill in students' curricula.

### 4.2.3. Between 2020–2023

The period 2020–2023 witnessed an influx in the number of studies on Arabic MT and a diversity in research frameworks and foci. Evaluative studies stressed the significance of translation evaluation to improve the performance of Arabic MT engines and suggest strategies to overcome relevant shortcomings [55,61,65]. Considering the limited research on Arabic MT effectiveness in dealing with specialized content [100], some studies addressed MT shortcomings in translating specialized texts or textual components such as the legal discourse [100], literary features [66], proverbs [67], sentiment words [101], relative clauses [57], as well as social media vernacular [102]. Evaluative studies continued to dominate the literature on MT systems to highlight their affordances and limitations in terms of adequacy, accuracy, fluency, context sensitivity, terminology and other criteria and called for complementarity between MT and HT via pre-editing and postediting [50,53,58,63,70,103]. However, they continued to overlook advanced quality criteria that require human critical thinking.

Perceptual studies conducted to evaluate professional translators' perceptions on using CAT tools and MT and practicing MTPE highlighted patterns of reluctance and avoidance of using MT technologies among translator communities in the Arab world [26,103]. This

result is compatible with [34] on the low popularity of postediting amongst the community of practicing translators. This result is significant, as it explains the lack of recognition of MT-related translator competencies like postediting. Ref. [71] concluded that Arabic translators avoided using MT during the pandemic despite the need for MT–HT integration due to software's lack of sensitivity to cultural and linguistic divergences. This indicates the value of perceptual studies in exploring new aspects of Arabic NMT system limitations.

Some studies investigated the topic from the perspective of translation learners and trainers. Ref. [56] researched building a sentence-aligned, error-tagged undergraduate translation learner corpus as an enriching resource for translation pedagogues and researchers. Ref. [104] drew a comparison between MT and HT in terms of adequacy and highlighted the value of MT for educators and professional translators. Ref. [25] tested the accuracy of NMT in rendering the linguistic features of tense and aspect to inform Arabic-into-English MTPE curricula and training.

Following the pandemic and the complete switch to online education, researchers in the field investigated the overuse of MT engines by translation learners and the impact of translation apps on trainee student performance. Some studies suggested the application of pre-editing strategies to improve MT output [62], while others stressed the necessity to integrate the use of translation apps into translation education [105]. Ref. [106] evaluated the potential of enhancing students' competency in MTPE. Although these studies were implemented within the framework of translation education, they did not tackle the topic empirically and directly, and therefore no explicit pedagogic implications for MTPE were provided (semi-pedagogic).

Despite the surge in MT research following online teaching, as shown in Table 5 and Figure 5, few studies adopted a pedagogic approach to the topic, providing explicit pedagogic implications for integrating MT technologies with translator education. Ref. [64] provided a description of a MT course focusing on translation students' postediting of free online MT output to improve translation student training and highlight the affordances of integrating technology into translation curricula. Ref. [28] accentuated the shift needed in English/Arabic translation pedagogy and curricula and evaluated the effectiveness of MTPE training in terms of productivity and quality. Ref. [27] experimented with the application of MT error identification to improve translation students' postediting practice. The research aimed to enhance the efficacy of translation training courses to keep pace with the continuous advancement in translation technologies and market needs. Table 6 below provides data collection results and taxonomy by research approach and methods for a clear representation of predominant trends and conceptual evolution in the literature.

Table 7 shows that the most common approach to Arabic MT research between 2000–2023 was the evaluative approach that used mixed methods (45%), followed by the empirical developer model, which adopted a technically oriented approach to the subject of research (25%). While technical studies adopted empirical methods, evaluative studies used mixed methods based on quantitative and qualitative data collection and analysis. Mixed methods are the most common in MT research, as they provide an objective evaluation of the product, considering their use of automated and human metrics. This result is in line with [11]. This implies that Arabic MT research was predominantly focused on experimenting with, evaluating and improving MT systems, with less attention paid to integrating these systems in the academic or professional practice of translators.

**Table 7.** Literature taxonomy by research approach and methods 2000–2023.

| Research Approach and Methods | Number | Percentage |
|---|---|---|
| Evaluative user model (mixed) | 27 | 45% |
| Empirical developer model (technical) | 15 | 25% |
| Translator training (mixed/empirical) | 8 | 13.3% |
| Survey (descriptive/qualitative) | 6 | 10% |
| Perceptual (quantitative) | 4 | 6.6% |
| Total | 60 | 100% |

Results show that there was an emerging but immature attention paid to translator practice and training frameworks. This is evident in the low percentage of perceptual studies (6.6%), surveys (10%) and studies conducted from the perspective of translator training (13.3%). The latter indicator covers semi-pedagogic and pedagogic studies. It is worth mentioning that the most appropriate research method during education transformation is the action research method, which is an empirical method that explores the experimentation of novel educational practices in response to shifts in educational contexts. This result is compatible with [29,31,32]. Only two studies in the literature adopted action research methods to the topic [27,28]. Figure 6 provides visual representation of the approaches and methods in the literature.

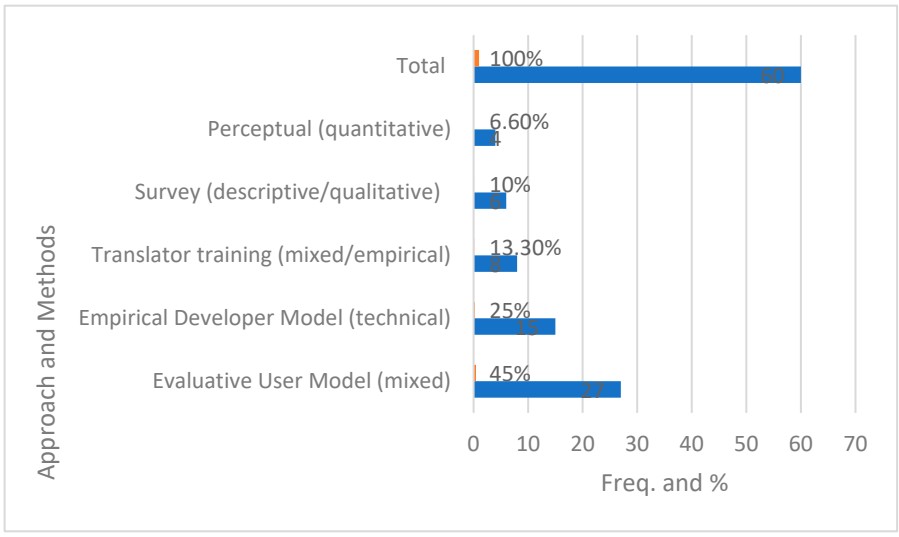

**Figure 6.** Approach and methods of the literature 2000-2023.

Upon comparing this result with the results in Section 1 on study types, the findings show that Arabic MT research funding has been directed to technical and evaluative studies, with less attention paid to studies related to professional or academic practices. This result is in conflict with the review conducted by [48] on the prominence of AI-empowered NMT and integrating human postediting as research topics that contribute to improving translation technology processes and performance. Evaluative and technical studies are a pillar in the improvement and mainstreaming of AI-powered MT practices [107], but these need to be complemented with more studies on MT–HT interaction and translation pedagogies and training [70]. The lack of studies on MTPE in translator training and education programs reflects the lack of technological integration with translation education in the Arab world and the need for incorporating responsive policies and practices in academic institutions.

## 5. Conclusions and Recommendations

Studies on Arabic MT over the last twenty-three years show a gradual progression in the status of MT on the academic and scholarly levels in the Arab world and continuous improvement in the performance of Arabic MT systems. However, there has been minor attention paid to HT involvement in the workflow of automatic translation systems. Academic and professional translator communities showed recognition of MTPE as a tool to improve and evaluate available Arabic MT systems, but there was little recognition on scholarly and educational platforms of postediting as part of the skillsets and digital literacies of translator training and education in the Arab world. This review shows that research funding and publication efforts on Arabic MT focused on technical aspects and MT software evaluation while neglecting trending topics that address translation education parameters such as integrating technology with translator education and introducing novel pedagogic practices that respond to the needs of twenty-first century translators [19,21,34,44,45,98,99].



Incorporating MTPE with the translation curricula should follow a gradual approach that addresses translation trainees' needs in accordance with their levels. Instructors of early-stage undergraduate translation students can use strategies that help learners with lower-level competencies such as identification, analysis and correction of MT errors which are normally of a linguistic nature, such as morphology, tense, semantics, etc. Progressively, translation pedagogues can develop learners' pre-editing competencies to simplify syntactic structures by using strategies of text segmentation and formatting and contextualizing lexical items to eliminate ambiguous references and content [44,88]. At advanced translation program stages, translation instructors need to develop the learners' critical thinking competencies [18], such as terminology management [36,44] and how to address deep translation issues related to MT processing of stylistic aspects, cross-cultural communication, as well as ethical and societal issues.

The findings of this systematic review signal a development in the status and uses of Arabic NMT and a transformation in relevant scholarly endeavours. However, there is underrepresentation of translation learners' competencies related to harnessing MT tools and systems and an urgent need to bridge the gap between translator education programs and labour market needs [47]. Therefore, the findings further suggest that one of the priorities for academic, scholarly and policymaking translation communities in the Arab world is to acknowledge MTPE as a high-order digital literacy to be developed gradually in translator training programs. MTPE should be part of the educational package provided in Arabic translation programs. There is a pressing need to conduct studies that provide an updated and deep comparative evaluation of available Arabic MT systems and inform this evaluation with postediting practices by human translators. This can be achieved by adopting empirical research approaches, such as action research methods and focusing on more advanced quality criteria in evaluating MT performance, such as ethical, cultural as well as societal limitations of NMT [40,82,83].

Also, it is important to fund research projects that lead to publishing books, book chapters as well as PhD studies [108] on the topic and to conduct more studies on Arabic NMT systems in Arabic–English translation. More research is required on Arabic MT of low-retrieval text types such as journalistic texts, culturally embedded texts, literary texts, online reviews and others. Future research on Arabic MT needs to incorporate the pedagogic practices and implications of MT applications in translator training and education.

**Author Contributions:** Conceptualization, L.I.O. and A.A.S.; methodology, L.I.O. and A.A.S.; software, L.I.O. and A.A.S.; validation, L.I.O. and A.A.S.; formal analysis, L.I.O. and A.A.S.; investigation, L.I.O. and A.A.S.; resources, L.I.O. and A.A.S.; data curation, L.I.O. and A.A.S.; writing—original draft preparation, L.I.O. and A.A.S.; writing—review and editing, L.I.O. and A.A.S.; visualization, L.I.O. and A.A.S.; supervision, L.I.O. and A.A.S.; project administration, L.I.O. and A.A.S.; funding acquisition, L.I.O. and A.A.S. All authors have read and agreed to the published version of the manuscript.

**Funding:** This research received no external funding.

**Institutional Review Board Statement:** Not applicable.

**Informed Consent Statement:** Not applicable.

**Data Availability Statement:** Data used in this study will be shared upon request.

**Conflicts of Interest:** The authors declare no conflict of interest.

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
