# Peer review of "Systematic Review of English/Arabic Machine Translation Postediting: Implications for AI Application in Translation Research and Pedagogy"

_informatics, doi:10.3390/informatics11020023_

Round 1

Reviewer 1 Report

Comments and Suggestions for Authors

The paper provides a systematic review on the state of research in English/Arabic machine translation. The PRISMA-based methodology allows for a rigorous and multidimensional synthesis of the literature, and the findings highlight key trends, gaps, and implications for the field. I suggest the following minor changes: 

  1. While the focus on English/Arabic MT is important and justified, perhaps include a brief discussion of how your findings relate to Arabic MT more generally. This could help broaden the relevance and impact of your work.
  2. Add a summary of key findings (as opposed to just approaches and methods) within English/Arabic MT, or at least a short synthesis of where to go from here.
  3. With regard to the previous point, what are the most pressing priorities and promising directions for researchers, educators, practitioners, and policymakers based on your findings?
Comments on the Quality of English Language

A final proofread will make the text better.

Author Response

  1. Summary

We thank you very much for taking the time to review this manuscript. You may kindly find the detailed responses below and the corresponding revisions/corrections highlighted/in track changes in the re-submitted files.

  1. Point-by-point response to reviewer’s comments and suggestions

Comments 1. The findings reveal a predominant emphasis on evaluating and improving machine translation software in English/Arabic postediting, with less emphasis on the skillsets and competencies of translators. While this study significantly contributes to the field of machine translation and postediting, it overlooks crucial aspects related to translators' skillsets and competencies (i.e., cultural understanding, time management, critical thinking, etc.), neglecting the human element in the postediting process and potentially compromising translation quality. It is in this sense that the study fails to adequately address the specific skills and competencies required for effective postediting in English/Arabic translation, limiting the efficacy of postediting efforts. The study also neglects to explore broader social impacts and pedagogical strategies for enhancing English/Arabic Machine Translation Postediting. Therefore, addressing these issues necessitates a comprehensive approach that considers both technological advancements in machine translation software and the development of translators' skills and competencies.

Response 1. Thank you for pointing these points out. We fully agree with these comments. Therefore, we have updated our discussion of the topic throughout the sections on results and findings from the perspective of translators’ competencies and skillsets. As for neglecting the human element in postediting processes, this is a dominant distinctive feature of the reviewed studies and a major finding of this systematic review. Therefore, to eliminate any ambiguity, and considering the significance of this finding, the authors elaborated on highlighting this gap in the reviewed literature in different parts (see pages 12, 16, and 21). Also, the authors pointed out the limitations of the reviewed literature in addressing social, ethical as well as cross-cultural communication issues, as indicated in track changes in the sections on discussion and findings in the revised manuscript. We hope you find these amendments pertinent.

Comments 2 on quality of the English Language: There are several doubtful statements:

  1. You say that NMT "employs artificial intelligence (AI)", while RBMT and SMT does not? It calls for a definition of AI.

Response 2 to point (a) under doubtful statements: Totally agree. A brief account on AI and its relation to NMT has been added to the introductory account on MT evolution in p. 2. The added content explains why NMT is AI-powered, as opposed to earlier MT models (RBMT and SMT). We hope the reviewer finds this revision relevant.

  1. "MTPE is becoming a popular topic in MT research," MTPE research exists since the 1980s.

Response 2 to point (b) under doubtful statements: agree with this point. The authors addressed this point by changing the statement to “MTPE continues to gain popularity in MT research”

  1. "When MT started to gain popularity at the beginning of the new millennium," it you are talking about the Arabic world, you should add that.

Response 2 to point (c) under doubtful statements: This is a valid comment. As suggested, the authors added “the Arab world” to the statement.

  1. Some typos, e.g.:

Krings (2001) Believed --> believed

Yanada --> Yamada

Response 2 to point (d): The authors acknowledge the need to revise the manuscript thoroughly by conducting careful proofreading of the manuscript’s content. Therefore, the paper has been revised meticulously and edited given the reviewer’s valuable remarks including aspects related to language, spelling, consistency in authors’ names in intext citations and list of references, punctuation, clarity, style, citation as well as referencing. We hope the amendments have contributed to an improved version of the research paper.

  1. Additional clarifications:

The authors would like to seize this opportunity to extend their utmost appreciation and recognition to the journal editors, editorial staff, as well as reviewers for their support following submission and during the review process. Your sincere efforts have contributed to improving the quality of this study and added value to our scholarly endeavour. Thank you once again for all your efforts, time, and support.

Reviewer 2 Report

Comments and Suggestions for Authors

This study delves into the domain of postediting in machine translation, with a specific focus on English to Arabic translation. It underscores the importance of postediting as a critical digital literacy to be cultivated among Arabic translation students, while also highlighting the need to bridge existing gaps in research and pedagogy within machine translation (MT) education. The research emphasizes the necessity of training translators in postediting machine-translated texts and stresses the imperative of enhancing postediting proficiency among Arabic translation students. This study utilizes the Preferred Reporting Items for Systematic Review and Meta-Analysis to examine 60 studies conducted since the early 21st century, aiming to identify prevailing trends and research gaps. The findings reveal a predominant emphasis on evaluating and improving machine translation software in English/Arabic postediting, with less emphasis on the skillsets and competencies of translators. While this study significantly contributes to the field of machine translation and postediting, it overlooks crucial aspects related to translators' skillsets and competencies (i.e., cultural understanding, time management, critical thinking, etc.), neglecting the human element in the postediting process and potentially compromising translation quality. It is in this sense that the study fails to adequately address the specific skills and competencies required for effective postediting in English/Arabic translation, limiting the efficacy of postediting efforts. The study also neglects to explore broader social impacts and pedagogical strategies for enhancing English/Arabic Machine Translation Postediting. Therefore, addressing these issues necessitates a comprehensive approach that considers both technological advancements in machine translation software and the development of translators' skills and competencies. Despite these shortcomings, this is a novel area of inquiry, not often explored by linguists and translation scholars. This novelty renders the study original and significant for both academic linguists and translation practitioners alike.

Originality: This paper is original as it contributes to the field of English/Arabic Machine Translation Postediting; it also provides insights onto AI Application in Translation Re-search and Pedagogy. The paper presents novel research findings, insights, or methodologies that contribute to the advancement of the field of informatics.

Theoretical Orientation: This study nicely reviews the literature on Arabic-English machine translation post editing and highlights not only the three forms of automatic machine translation but also sheds light on digital literacies, translator training, and machine translation qualities, capabilities and impediments. The study also makes a substantive contribution to the field of informatics by extending existing knowledge, proposing some new theories or models, or presenting practical applications with real-world implications.

Research Design: The author uses the “Preferred Reporting Items” for Systematic Review and Meta-Analysis and provides a quantitatively informed synthesis and analysis of the relevant literature by extracting and interpreting data from published research on the topic. The aim is to provide evidence and critical interpretation of dominant research trends and inform future research and practices in the field of Arabic/English MT. Overall, the research methodology used in the paper is rigorous and sound. It adheres to appropriate research design, data collection, analysis, and interpretation techniques, ensuring the validity and reliability of the findings.

Language: The language of the paper is solid, although there are some language issues and misuses of punctuation marks. The paper is well-written, organized, and structured, with clear and concise presentation of ideas. It effectively communicates the research objectives, methods, results, and conclusions to the target audience.

Concluding Remarks: It is clear that this study has good merits. Therefore, it meets the standards of publication in Informatics. The study presents novel research findings, insights, or methodologies that contribute to the advancement of the field of informatics. The work is original as it addresses significant research questions or problems within the domain of machine translation and post-editing. The study also demonstrates relevance to current trends in machine translation and highlights the challenges, or gaps in knowledge within this field. With a little bit of revision, it is my opinion that the paper is suitable for publication in the Informatic Journal.

Comments on the Quality of English Language

There are several doubtful statements:

You say that NMT "employs artificial intelligence (AI)", while RBMT and SMT does not?  It calls for a definition of AI.

"MTPE is becoming a popular topic in MT research," MTPE research exists since the 1980s.

"When MT started to gain popularity at the beginning of the new millennium," it you are talking about the Arabic world, you should add that. 

Some typos, e.g.:

Krings (2001) Believed --> believed
Yanada --> Yamada

Author Response

  1. Summary

We thank you very much for taking the time to review this manuscript. You may kindly find the detailed responses below and the corresponding revisions/corrections highlighted/in track changes in the re-submitted files.

  1. Point-by-point response to reviewer’s comments and suggestions

Comments 1. While the focus on English/Arabic MT is important and justified, perhaps include a brief discussion of how your findings relate to Arabic MT more generally. This could help broaden the relevance and impact of your work.

Response 1. Thank you for pointing this out. We fully agree with this comment. Therefore, we have updated our discussion of the findings accordingly. Also, we have added a brief account to the section on findings to explain how they broadly relate to Arabic MT research. All changes have been marked in track changes in the revised manuscript. We hope you find these amendments pertinent.

Comments 2 & 3. Add a summary of key findings (as opposed to just approaches and methods) within English/Arabic MT, or at least a short synthesis of where to go from here. Concerning the previous point, what are the most pressing priorities and promising directions for researchers, educators, practitioners, and policymakers based on your findings?

Response 2 and 3. This is a valid point which needs to be considered to enhance the study’s scholarly impact and contribution. The findings’ section has been revised in light of this valuable remark. We have provided a synthesis of the study’s findings vis-a-vis earlier scholarly contributions to the topic from the Arab world and beyond. Also, we have highlighted the main priorities and future directions for all involved parties stressing the significance of introducing new quality parameters to research on evaluating MT engines, integrating technology with translator education in higher education institutions, as well as adopting empirical research methods that bridge the gap in research endeavours between human translation and machine translation. All amendments are marked in track changes in the edited manuscript. Hopefully, these amendments have succeeded in addressing this important point of concern.

Comment 4 on the Quality of English Language: A final proofread will make the text better.

Response 4: The authors acknowledge the need to improve the study’s quality of the English language by conducting a final proofread of the manuscript’s content. Therefore, the paper has been thoroughly revised and edited in terms of language, punctuation, clarity, style, citation as well as referencing. We hope the amendments have contributed to an improved version of the research paper.

  1. Additional clarifications:

The authors would like to seize this opportunity to extend their utmost appreciation and recognition to the journal editors, editorial staff, as well as reviewers for their support following submission and during the review process. Your sincere efforts have contributed to improving the quality of this study and added value to our scholarly endeavour. Thank you once again for all your efforts, time, and support.